# Traditional Village Landscape Integration Based on Social Network Analysis: A Case Study of the Yuan River Basin in South-Western China

**Daimou Wei** [1,2], **Zhexiao Wang** [1,2] **and Bin Zhang** [1,2,*]

1    Department of Landscape Architecture, College of Horticulture and Forestry, Huazhong Agricultural University, Wuhan 430070, China; weidm@webmail.hzau.edu.cn (D.W.); welixchina@gmail.com (Z.W.)
2    Key Laboratory of Urban Agriculture in Central China, Ministry of Agriculture and Rural Affairs, Wuhan 430070, China
*    Correspondence: zhangbin@mail.hzau.edu.cn

**Abstract:** Traditional rural Chinese landscapes have fragmented from the impact of rapid urbanisation and modernisation. Aiming to address this tough issue, the Chinese central government proposed the Traditional Villages Project, which is top-down traditional village management and conservation policy. A traditional village landscape network (TVLN) can be used to integrate rural landscapes and ensure the unified protection of natural and cultural landscapes. This paper aimed to establish a method of building a TVLN through three main steps: the calculation of the connection strength of traditional villages, calculation of the tie strength between traditional villages, and establishment of a TVLN. The results demonstrated the rich layers and stable structure of the Yuan River Basin's TVLN, but there was a hidden risk in its stability due to the existence of tangent and isolated points. This TVLN quantitatively examined the characteristics and relationships of traditional villages and provided data support for the approval of traditional villages and protection policy formulation. A TVLN can support the overall conservation of traditional village landscapes, enhance their comprehensive value, and promote the sustainable management and cross-regional protection of traditional village landscapes.

**Keywords:** landscape heritage; connection strength; social network analysis; rural landscape; sustainable protection

## 1. Introduction

The products and living activities of aborigines have created a unique traditional village landscape [1,2] that reflects the wisdom of humans surviving in a natural environment. As a cultural landscape, traditional villages are the result of long-term interactions between humans and nature in a certain area [3]. Traditional village landscapes are important elements of a country's or region's character [4,5]. They must be protected to maintain national identity and culture and ensure sustainable development in rural areas [6].

China has a history of more than 7000 years of agricultural civilisation. Traditional Chinese villages carry rich historical, ethnic, and regional cultural information [7]. They are basic social units built to meet the needs of production and life through agricultural civilisation [8]. Since 1978, China has experienced unprecedented modernisation and urbanisation [9], which has led to decreases in the rural population and a loss of vitality. From 1986 to 2020, the number of natural villages in China decreased from 3.65 million to 2.36 million, and the number of administrative villages decreased from 0.85 million in 1986 to 0.49 million [10,11]. The rural population has also decreased from 82.1% in 1978 to 39.2% in 2020 [12,13]. The large-scale construction of modern residential buildings has impacted the traditional patterns of village landscapes, such as their colours, shapes, and textures [14]. Many vernacular buildings have been abandoned, and traditional

village patterns have become fragmented [15,16]. The ecological environment and cultural diversity of traditional villages have also gradually deteriorated [17,18]. As China's rural areas undergo transformation and reconstruction due to urban sprawl, the old urban-rural relationship begins to disintegrate [19–21].

In 2012, the Chinese government proposed the Traditional Villages Project, which covers 6819 traditional villages according to the indicators such as the longevity, scarcity, richness and integrity of architectural forms, the value of craft aesthetics, and the heritage value of craft and construction, including 307 in Yuan River Basin. The Yuan river is one of the four major water systems in the Wuling mountain area. It is the area with the most advantageous natural and living conditions in the Wuling mountain area. The Wuling mountain area is located in Southwest China and spans four provinces (cities) of Hunan, Hubei, Guizhou, and Chongqing. Due to its mountainous location and inconvenient transportation, the Wuling mountain area is less affected by the development of urbanisation and modernisation, and the traditional villages are well preserved. The number of selected traditional villages in the Yuan River Basin has increased sharply batch by batch, indicating that the region is rich in traditional settlement landscape resources and protection advantages. With its implementation, traditional village protection has fascinated scholars and planners for some time [22]. At present, research on the topic mainly focuses on spatial forms, vernacular architecture [23], heritage value [24], protection mechanisms and criteria [25], and sustainable protection [26–29]. Sustainable protection emphasises the integration of the natural environment, cultural context, and creators of traditional villages. Protection that is integrated with cross-regional and cross-cultural heritage has become an international consensus, and the scope of protection has shifted from settlement to corridor [30,31]. Corridor scale networks are becoming increasingly important for the sustainable development management of society, economy, and culture [32,33]. Since traditional villages are formed in the same geographical environment with similar economic, cultural, and ecological backgrounds, the construction of landscape networks can be conducive to their sustainable management and protection.

To date, research on traditional village landscape networks (TVLNs) has focused on qualitative methods more than quantitative research [34,35]. This study proposes a method to study TVLNs based on social network analysis (SNA), a sociological tool used to quantify the relationship between actors and the role of social actors. To the best of our knowledge, this is the first study to use SNA to create a TVLN. TVLNs quantify the internal relationship between traditional villages, connecting their regional-scale protection planning with the settlement-scale protection schemes, which provides data support for the sustainable management and protection of villages.

## 2. Materials and Methods

### 2.1. Study Area

The Yuan River Basin is the largest river system in the Wuling Mountain area. We used the ArcGIS hydrological tool to examine digital elevation model (DEM) data for the Wuling Mountain area and obtained Yuan River, Qing River, Wu River, and Li River systems. The county-level government is the basic unit of official statistics [36]. For the convenience of data acquisition, this study was based on the Yuan River Basin, and the study area was defined by modifying the county boundary within the basin (Figure 1). The study area was approximately 34,500 km$^2$, including 16 complete and 5 incomplete counties.

The Yuan River Basin has an astonishingly diverse landscape, which can be explained by two main factors. First, it has diverse natural conditions. The Yuan River Basin has a subtropical climate. It is located between 26°–30° N latitude and 107°–112° E longitude and has an altitude of 105–2350 m. It has abundant light, heat, and water sources during the four distinct seasons. The Yuan River Basin has a typical karst landform with mountains, hills, flat dams, and plains. The unique topography and landforms produce rich landscape resources and biodiversity. Second, the Yuan River Basin has a long history of human settlement. It has historically been the main channel of multi-ethnic commercial and

cultural exchanges. Since the 1280s, the Yuan River Basin has been an autonomous region ruled by hereditary leaders of ethnic minorities. After a long struggle with the central government, a large number of military settlements and cultural lines were formed. In 1735, the rule system of hereditary leaders of ethnic minorities was abolished, and the central government appointed rotating officials. An immigration policy was also implemented. With the rapid growth of the population, settlement construction gradually developed. In the long-term struggle and integration of the Han, Dong, Gelao, Miao, and Tujia, a rich cultural landscape was created. The Yuan River Basin lies in a mountainous area with inconvenient transportation. Since the 20th century, traditional transportation has been gradually replaced due to the construction of expressways and railways. The construction of water conservancy facilities led to the suspension of channels, accelerated the decline of traditional traffic routes, and fragmented the landscape pattern formed by traffic routes. Urbanisation and modernisation in the area are slow, and the villages have maintained a relatively historical style. To date, 307 traditional villages have been selected for their historical, cultural, scientific, artistic, economic, and social values by the government in the Yuan River Basin, which indicates that they have abundant landscape resources and good protection conditions in the area.

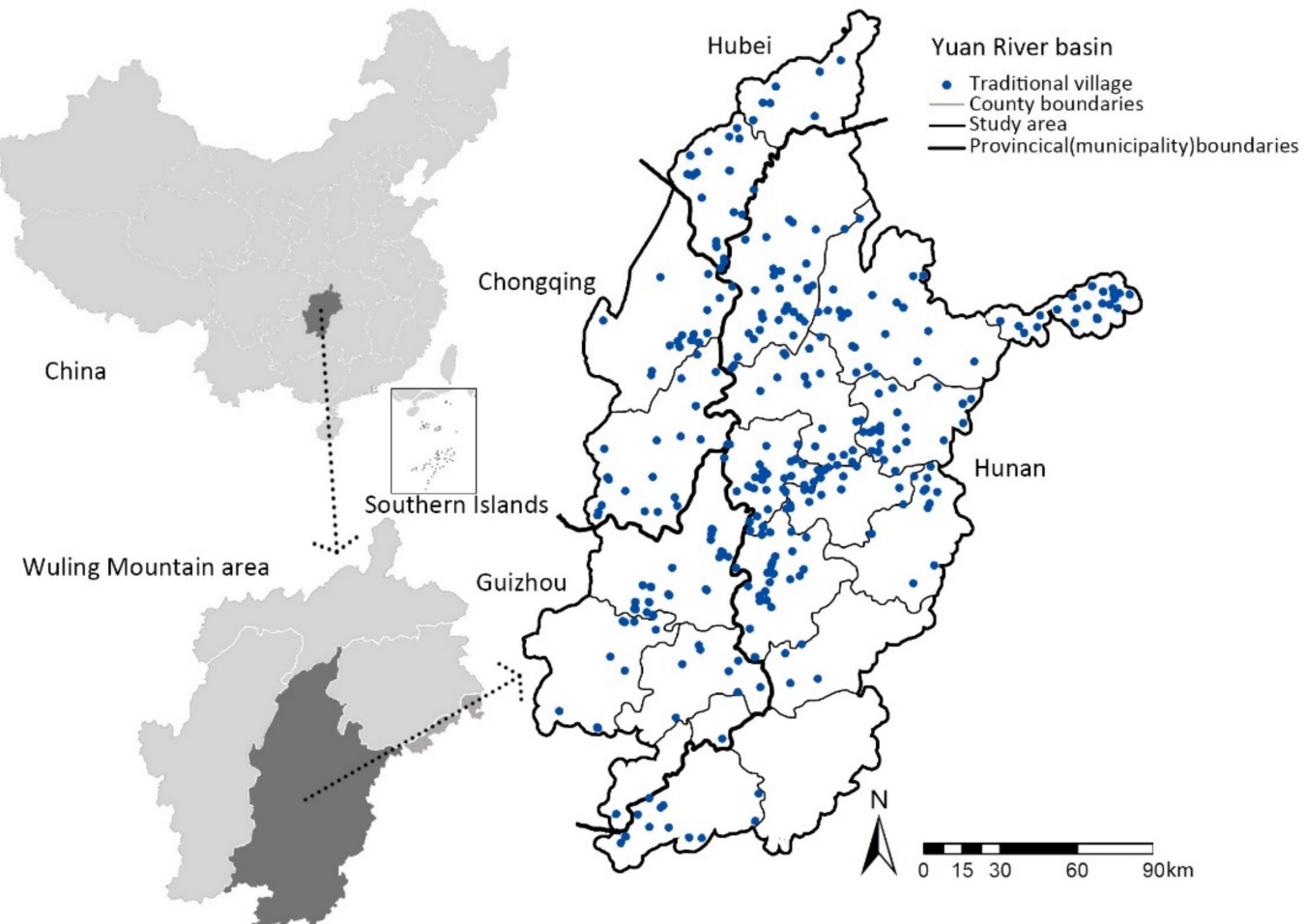

**Figure 1.** Location of the Yuan River Basin and distribution of traditional villages.

### 2.2. Establishing a TVLN

The construction of a TVLN involves three main steps: (1) calculating the connection strength of traditional villages, (2) calculating the tie strength between traditional villages, and (3) establishing a TVLN.

### 2.2.1. Connection Strength

Connection strength is the attraction between traditional villages based on their own resource endowment and the possibility of forming village clusters with it as the core. In China, the formation of traditional villages is related to the geographical environment [37,38], agricultural production mode [39,40], inheritance mode, and historical and cultural development [41–43], which can be divided into national aggregation, kinship aggregation, complementary aggregation of production mode, and sharing aggregation of advantageous natural factors. Based on the above, 15 factors were selected and divided into four categories: economy, geography, culture, and ecology (Table 1). A database comprising a vector view of traditional villages, buildings, farmlands, and linear heritage was established based on Google Earth WGS 1984; this database is summarised in Table 2. Based on SPSS (version 22.0) correlation analysis, this study employed the Analytic Hierarchy Process (AHP) to assign the weights for the evaluation factors of the connection degrees of traditional villages (Table 1). After dimensionless processing of the connection degree factor values of traditional villages, the connection degree of traditional villages was calculated according to the weight of the factors.

**Table 1.** Connection strength and weight factors.

| Categories | Factors | Weight |
|---|---|---|
| Economy | GDP | 0.0773 |
| | Population density | 0.0712 |
| | Farmland productivity potential | 0.0380 |
| | Per capita construction land | 0.0306 |
| | Per capita farmland | 0.0245 |
| Geography | Core density of traditional villages' POI | 0.0789 |
| | Core density of linear heritage | 0.0789 |
| | Point–point proximity | 0.0245 |
| | Point–line proximity | 0.0425 |
| Culture | Degree of multi-ethnicity | 0.0504 |
| | Land fragmentation index | 0.2634 |
| | Extension degrees of building groups | 0.0575 |
| | Finger-like degrees of building groups | 0.0447 |
| | Construction–agriculture geological distance | 0.0504 |
| Ecology | Ecological sensitivity | 0.0672 |

GDP, population density, farmland productivity potential, per capita agricultural land, and per capita construction land are the economic factors. GDP is the total output of a region during a certain period. It is recognised as the best indicator of the national economic situation. People are an important productivity resource and provide consumption markets. Population density is an important indicator of regional economic situations. Agriculture is the main industry in traditional villages. Farmland productivity reflects the productivity resources of rural areas. Farmland productivity potential was calculated according to the distribution, soil, and elevation of the agricultural land. By employing the Jenks tool in ArcGIS, the potential productivity of farmland was divided into five levels. Per capita agricultural land and per capita construction land were calculated according to the total cultivated land area, total construction land area, and population of traditional villages.

The geographical factors include the core density of traditional villages' points of interest (POIs), the core density of the linear heritage, point–point proximity, and point–line proximity. The core density of traditional villages' POIs was analysed at 250 pixels of 30 km bandwidth based on the spatial data of traditional villages, which were divided into

9 levels. We selected the Sichuan Ancient Salt Routes, the fairway, the Miao Great Wall, and Zhi-Liu Railway as linear heritage. The core density of linear heritage was analysed at 250 pixels of 15 km bandwidth, which was divided into 8 levels. The point–point proximity analysis and point–line proximity analysis of ArcGIS were used to obtain the nearest traditional village and river system.

**Table 2.** Summary of variables and data.

| Data | Source | Units | Time |
|---|---|---|---|
| GDP | Geographic Data Sharing Infrastructure, College of Urban and Environmental Science, Peking University | RMB 10,000/sq. km | |
| DEM | Google Earth open access data | 30 m × 30 m | 2019 |
| Satellite imagery | Google Earth open access data | | 2019 |
| Land cover | European Space Agency (http://maps.elie.ucl.ac.be/CCI/viewer/index.php (accessed on 13 January 2019)) | 300 m × 300 m | 2015 |
| Spatial point data of traditional villages | Constructed based on Google Earth WGS1984 satellite image | | 2018 |
| Farmland productivity potential | Resource and Environment Science and Data Centre (http://www.resdc.cn/ (accessed on 23 July 2017)) | Kg/ha | 2017 |
| Population | National Bureau of Statistics (http://www.stats.gov.cn/tjsj/pcsj/rkpc/6rp/indexch.htm (accessed on 19 January 2020)) | | 2010 |
| Hydrological data | Resource and Environment Science and Data Center (http://www.resdc.cn/ (accessed on 20 February 2018)) | | 2007 |
| Administrative boundaries | Resource and Environment Science and Data Center (http://www.resdc.cn/ (accessed on 8 November 2018)) | | 2015 |
| Population density | Geographical Information Monitoring Cloud Platform (http://www.dsac.cn/ (accessed on 19 January 2020)) | person/sq. km | 2015 |

The cultural factor is comprised of degree of multi-ethnicity, land fragmentation index, extension degrees of the building group, finger-like degrees of the building group, and construction–agriculture geological distance. The degree of multi-ethnicity demonstrates the possibility of integrating the production and lifestyles of different ethnic groups. There were three categories for the landscape types of ethnic minorities in the study area: single ethnic group, dual ethnic group, and multi-ethnic group, with values of 1, 2, and 3, respectively. The land fragmentation index evaluates the human–land relationship in the traditional villages through the number of patches, the average area of patches, the standard deviation of the patch area, and patch aggregation degree. The larger the value, the stronger the mutual attraction between building clusters; thus, the higher the connection strength between traditional villages. The extension and index degrees of the building group are landscape pattern indicators used to judge the form of the building group. The formula calculation results based on the spatial data of the traditional village building patches were obtained using ArcGIS. The construction–agriculture geological distance represents the spatial relationship between the land on which the village was built and the land used for agriculture. Higher values for fragmentation of construction land, the extension of the construction group, distance from the agricultural, geological centre, and degree of finger were more conducive to supplying a traditional village with public service facilities and agricultural production activities.

Ecological sensitivity also affects connections between traditional villages. We superimposed the sensitivity of elevation, relief, slope, and land cover to obtain comprehensive ecological sensitivity.

### 2.2.2. Tie Strength

Tie strength refers to the degree of connection between two nodes, which is the comprehensive result of the degree of connection and distance attenuation effect. The tie strength *F* was determined as

$$F = \frac{m_i m_j}{d^2}$$

where *F* denotes the strength of the relationship between nodes *i* and *j*; $m_i$ and $m_j$ are the connection strengths of nodes *i* and *j*, respectively, which are determined by the resource endowment of the node itself; and d is the attenuation distance. The spatial straight-line distance between two nodes was taken as the attenuation distance, which was obtained using the Point Distance (Analysis) tool for neighborhoods in ArcGIS.

### 2.2.3. Establishment of Network

By taking 307 select traditional villages as nodes and the top 3% of the relationship strength as links, UCINET (version 6.0) was used to construct the TVLN of the Yuan River Basin.

### 2.3. Properties of TVLN Analysis

This section describes a system for assessing three different aspects of the TVLN: overall (tangent point and lambda set), partial (cohesive subgroup), and individual (centrality).

### 2.3.1. Tangent Point

The tangent point refers to the only intermediary connecting the internal structure of the network and is the network's weakness. There are two types of tangent points: those connecting two clusters and those connecting two nodes. The influence of the first type on network stability was stronger than that of the second type.

### 2.3.2. Lambda Set

The lambda set evaluates the factor of network hierarchy stability. It reflects the ability of the network to maintain connectivity to some nodes when their relationship is broken by measuring the 'border connectivity' index of two nodes in the network.

### 2.3.3. Cohesive Subgroup

The cohesive subgroup is the result of node clustering. Faction is a cohesive subgroup based on reciprocity, which is the largest complete subgraph with at least three points. In this study, we do not consider factional overlap; each node belongs to only one subgroup.

### 2.3.4. Degree Centrality

Degree centrality represents the uniform distribution of nodes. The higher the degree of centrality, the more nodes are directly connected to the node, and the more obvious the aggregation degree of the nodes in the network to the core nodes. These variables can be expressed as

$$C_{AD} = \frac{\sum_{i=1}^{n}(C_{ADmax} - C_{ADi})}{n^2 - 3n + 2}$$

where $C_{ADi}$ is the degree of node *i*, $C_{ADmax}$ denotes the maximum degree of network nodes, and *n* denotes the degree of the node.

### 2.3.5. Betweenness Centrality

The betweenness centrality indicates the extent of the equilibrium distribution of the network; the higher its value, the more time it plays a connecting role between other nodes.

In this study, traditional villages with a high betweenness centrality can communicate with multiple villages and form clusters through these nodes. The calculation formula is

$$C_B = \frac{\sum_{i=1}^{n}(C_{RBmax} - C_{RBi})}{n-1}$$

where $C_{RBmax}$ is the maximum betweenness centrality of all nodes, $C_{RBi}$ denotes the betweenness centrality of node *i*, and *n* is the degree of the node.

## 3. Results

### 3.1. The TVLN of the Yuan River Basin

There were 307 nodes and 2820 lines in the network, which was created by UCINET. Affected by the distribution of traditional villages in the space, the network topology presented a multi-centre and arborisation structure.

### 3.2. The TVLN's Characteristics

#### 3.2.1. Tangent Point

Twenty-one tangent points (6.84%) were identified in the network, including 14 of the first type and 7 of the second type. The first type of tangent point indicates that there are closely related groups in the network, which may be managed and protected by partial cooperation. The second type of tangent point indicates that the local structure of the network is loose.

#### 3.2.2. Lambda Set

Figure 2 shows the results of the lambda analysis. As shown in Figure 3, there are 29 edge-relatedness levels in the network, with a maximum value of 30 and a minimum value of 0. The proportion of nodes with an edge correlation degree value >17 is less than 2%, and the proportion with an edge correlation degree value ≤17 is between 2% and 6%, except for 1, 3, and 4. The ratio difference between the maximum and minimum of the edge-relatedness levels is 3.26%, which indicates that the network has a rich hierarchy and stable structure. The peak values of edge-relatedness values are 4 and 14. This indicates that there are two kinds of clusters with strong and weak correlations among individuals in the network, and the connections between them exhibit two extreme states of strong correlation and very weak correlation.

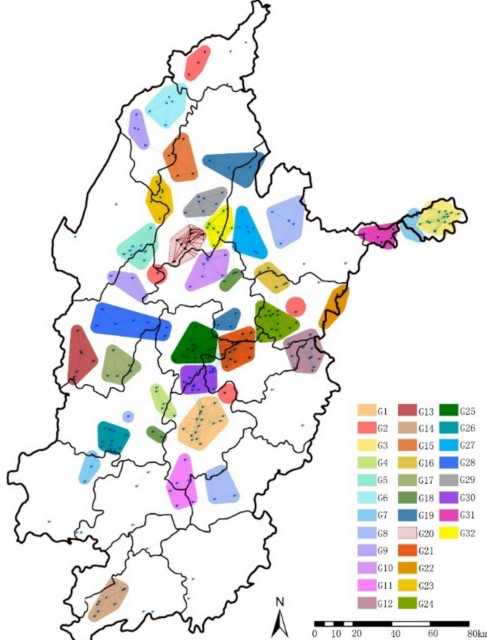

**Figure 2.** Distribution of levels of edge-relatedness.

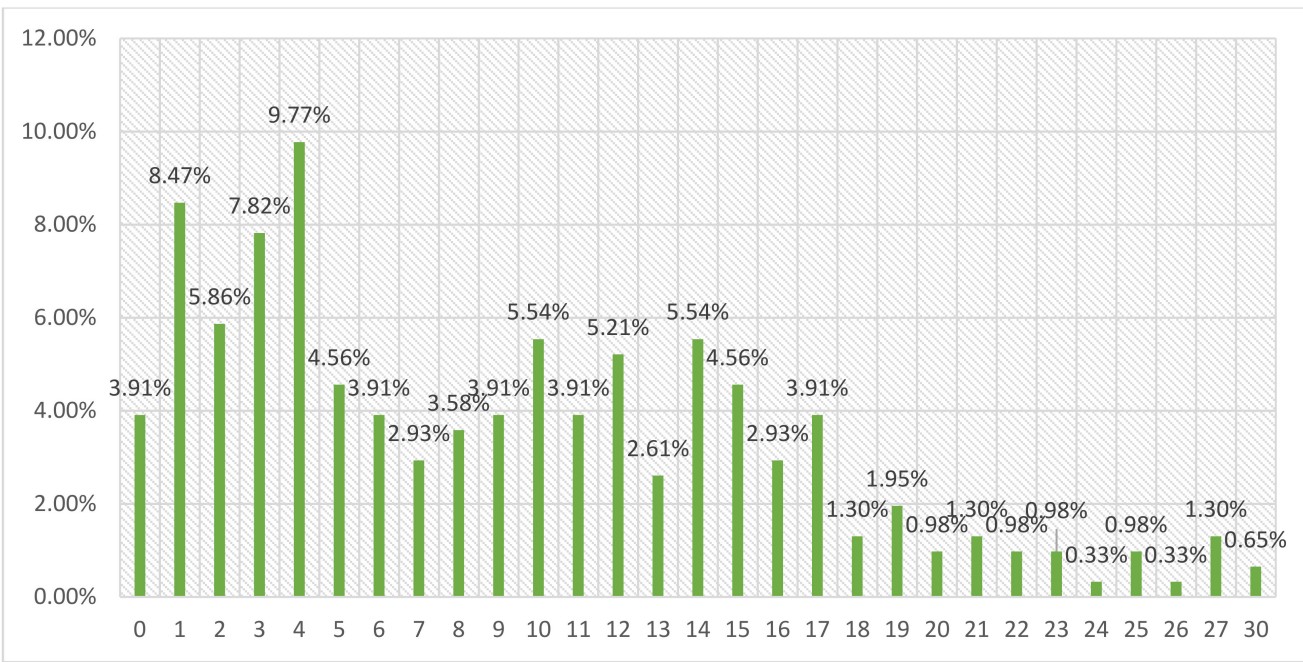

**Figure 3.** The spatial distribution of cohesive subgroup.

### 3.2.3. Cohesive Subgroup

Figure 4 shows the spatial distribution of cliques in the Yuan River Basin. When fitness is a valley value, 33 groups are obtained, including a scattered group composed of isolated points. The largest subgroup contained 18 nodes, the smallest subgroup contained 5 nodes, 5 subgroups contained more than 15 nodes, and 6 subgroups contained 5 nodes. There are 4 subgroups that included 3 county nodes and 13 subgroups that included 2 county nodes.

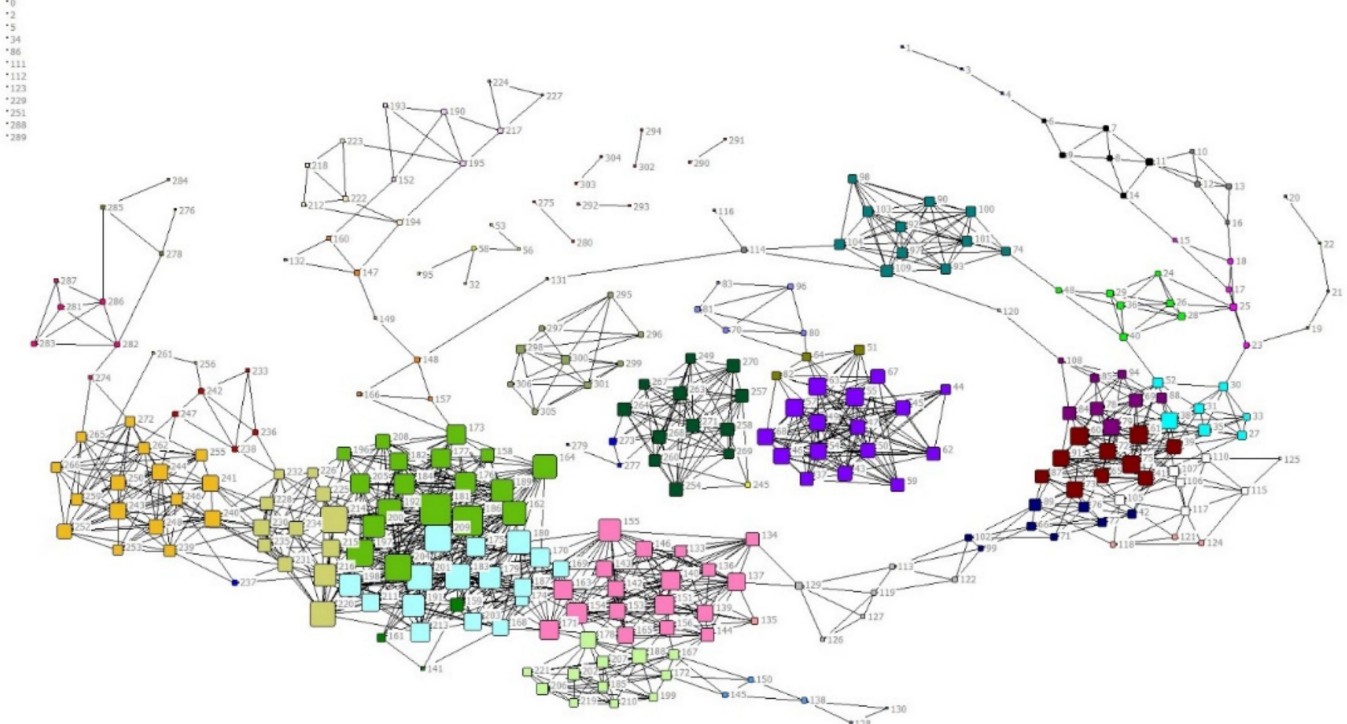

**Figure 4.** The degree centrality of the TLVN of the Yuan River Basin.

### 3.2.4. Degree Centrality

As shown in Figure 5, the size of a dot represents the degree centrality value. The node with the largest degree centrality is node x118, with a value of 31; except for isolated points, 19 nodes have a degree centrality of 1. The greater the difference in the degree centrality of nodes, the greater the degree centrality potential of the graph. The degree centrality potential of this study is 7.18%, and the overall centrality potential is more balanced. Owing to the negative correlation between the distance between the nodes and the relationship strength, the traditional village nodes with a high degree of centrality have a clear aggregation trend.

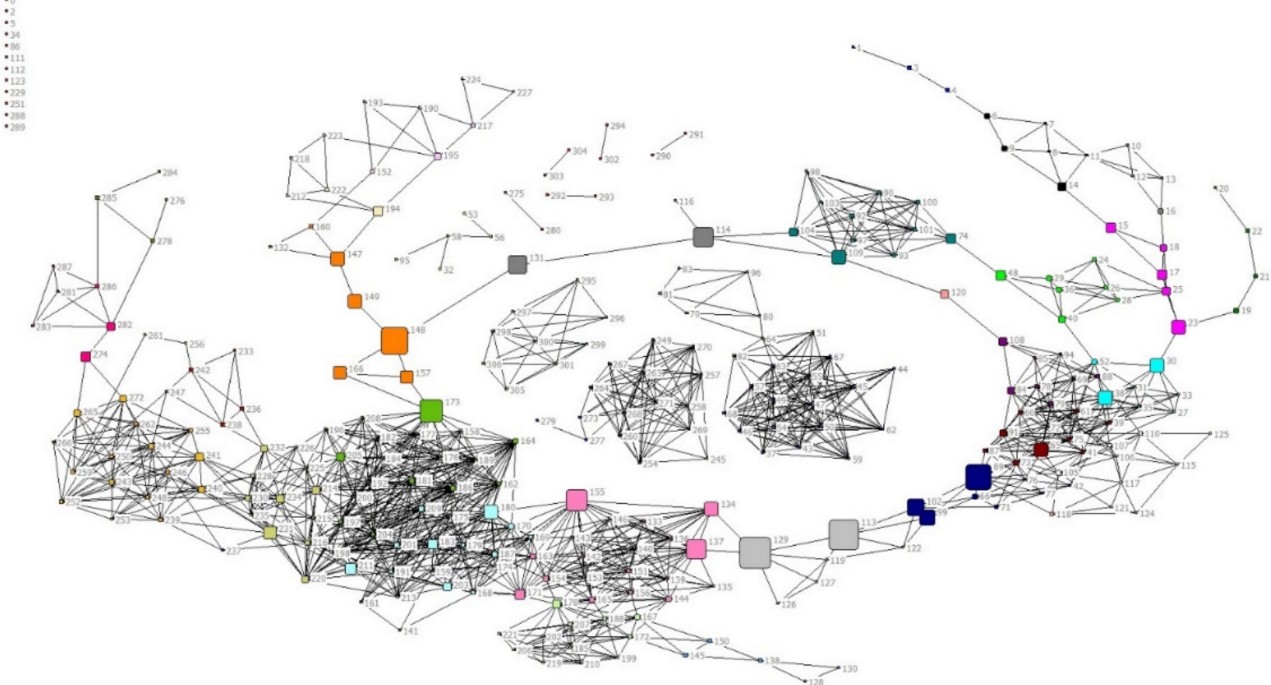

**Figure 5.** The betweenness centrality of the TLVN of the Yuan River Basin.

### 3.2.5. Betweenness Centrality

The size of the dot represents the middle centrality value (Figure 5). The node with the largest middle centrality is x73, with a value of 17.194. There were seven nodes with intermediate centrality $\geq 10$ and 16 nodes with intermediate centrality $\geq 5$. The middle centre potential of the network is 15.90%, which indicates that although the traditional villages in the network were divided into multiple groups, they are connected through multiple nodes, and there is a certain balance between the villages in the network.

## 4. Discussion

### 4.1. Describing the TVLN

The principle of practical adaptability in statistical mechanics [44] shows that the network characteristics of traditional villages in the Yuan River Basin are closely related to the spatial distribution and geographical characteristics of traditional villages. The landscape network of traditional villages in the Yuan River Basin is distributed in a tree shape with distinct clusters (Figure 2), which results from its topographic features and the uneven spatial distribution of traditional villages.

The distribution of cohesive subgroups is significantly related to the distribution of cultural routes. Two supergroups were formed in Fenghuang, Huayuan, and Jishou in the central and southern parts along the Miao Great Wall and the Sichuan Ancient Salt Routes, and in Longshan, Baojing, and Youyang along the Sichuan Ancient Salt Routes and waterways in the central and northern parts. Groups 4, 12, and 15 were distributed

along the Sichuan Ancient Salt Routes in Longshan and Changde. Groups 3, 6, 12, and 18 were distributed along the upstream channel of the You River. Groups 1, 8, 17, and 25 were distributed around the ancient Sichuan Ancient Salt Routes (Huayuan–Jishou–Fenghuang), Zhi-Liu railway, and Miao Great Wall. There were some independent, complete, dense subgroups distributed at the edge of the study area, such as Groups 2, 11, and 21.

### 4.2. Management and Protection of Traditional Villages

### 4.2.1. Grading Protection

The tangent point is the only local connection of the network and the key node that affects its stability. As described in the results in Section 3.2.1, traditional villages at the tangent point can be listed as core protection objects. The stability of the network can be improved by strengthening the construction of tangent points and setting parallel nodes. The isolated point is an unstable factor in the network. Most of the isolated points are traditional villages with a low degree of connection, located at the edge of the study area, and sparsely distributed. Isolated points can be designated as key protected objects. The corresponding protection plan is formulated according to the factors of connection strength of traditional villages located at isolated points. For example, nodes with low connection strength due to a long distance can build a local network by cultivating their adjacent nodes; nodes with low connection strength caused by economic backwardness can improve their connection strength and link to the network by supporting the characteristic economy and enriching industrial structure.

### 4.2.2. Protection of Traditional Village Clusters

A cluster is an ecological concept that refers to a structural unit that describes the cohabitation of a variety of species within a particular area or environment [45]. The protection of traditional village clusters refers to the purpose of complementary advantages and coordinated development by exploring the characteristics of traditional villages and integrating resources in a certain area. The cluster management and protection of traditional villages based on the cohesive subgroup mainly include the following two aspects. Firstly, it provides a basis for collaborative management and the protection of traditional villages across administrative regions. The county government is the smallest unit of national economic and social statistics and the most basic administrative unit that implements national and local policies. The linear distance between Longshan County and Laifeng County is 4 km, separated only by the You River. The traditional villages of the two counties obviously have much in common culturally and are ecologically interconnected. However, the two counties belong to different provinces, and there are barriers to the coordinated development of their traditional villages. Cohesive subgroup analysis found that traditional villages e1, e2, and e13 in Laifeng County and traditional villages x17 and x18 in Longshan County belong to the same subgroup, making them suitable for collaborative development in terms of improving the continuity of traditional village landscapes. Six subgroups included traditional villages in multiple cities, and 15 subgroups included traditional villages in multiple counties in the Yuan River Basin. Secondly, cohesive subgroups established a link between the regional scale of traditional villages and settlement-scale management and protection, which provides data support for regional-scale management and protection planning, as well as guidance strategies for settlement scale management and protection.

### 4.3. Protecting the TVLN

In order to protect the Yuan River Basin's TVLN, the network structure is optimised by strengthening the advantage nodes, focusing on the key nodes, and reducing the disadvantage nodes. Nodes with high centrality usually have excellent resource endowments and are located in the local centre of the network. The corresponding protection schemes can be formulated based on the resource characteristics of the nodes. For example, node x118, which has the highest degree of centrality, belongs to a typical karst terrain with a large

number of natural caves and peculiarly shaped mountains that are very rich in natural landscape resources. The node is a Miao village. The villagers there eat a Miao diet and engage in traditional folk activities. The highest node in the middle centre, x44, is a Tujia village, located on the bank of You River close to the wharf, with dangerous terrain. After the land reform in 1727, it became an important commercial town because of its developed transportation. The tangent point is the key node of the network and the key factor of network stability. Thus, the existence of a tangent point should be reduced. Outliers are weak nodes in the network and are mostly located in areas with sparsely distributed traditional villages. In the future, such an area can be prioritised in the application of traditional villages. At the same time, isolated points often have advantages in local areas, which can become the centre of local areas and form traditional village clusters.

## 5. Conclusions

This study proposes a method to investigate the connection between villages based on SNA, which can integrate and manage the village landscapes and promote the sustainable protection of traditional village landscapes. We analysed the top 3% of the tie strength constituted by the selected network. It was found that the landscape network of traditional villages in the Yuan River Basin has a dendritic and multi-core structure, which is closely related to the geographical characteristics of the Yuan River Basin and the distribution of traditional villages. Traditional villages located at key locations, such as the network's tangent point and isolated point, were identified, and corresponding protection strategies were proposed. This study only examined select traditional villages; there were many villages in the study area that were not selected but had excellent landscape value. In addition, only 3% of the network density was examined, but this did not change the properties of the network. Future studies can build multi-valued networks with different tie strength thresholds and analyse the characteristics of networks from different network densities. Moreover, by analysing the network characteristics formed by different batches of traditional villages and studying the dynamic changes of the network, suggestions are provided for the examination and approval of traditional villages in the future.

**Author Contributions:** D.W. contributed to conceptualisation, methodology, software, validation, formal analysis, data curation, and writing—original draft preparation. Z.W. contributed to the data collection, software, validation, data curation. B.Z. contributed to conceptualization, methodology, supervision, and writing—review and editing. All authors have read and agreed to the published version of the manuscript.

**Funding:** This work was supported by the National Natural Science Foundation of China (Grant No. 51678269) and the Chinese Fundamental Research Funds for the Central Universities (No: 2662018YJ017).

**Institutional Review Board Statement:** Not applicable.

**Informed Consent Statement:** Not applicable.

**Data Availability Statement:** Not applicable.

**Conflicts of Interest:** The authors declare no conflict of interest.

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
