# Peer review of "Traditional Village Landscape Integration Based on Social Network Analysis: A Case Study of the Yuan River Basin in South-Western China"

_sustainability, doi:10.3390/su132313319_

Round 1
Reviewer 1 Report
Dear colleagues,
You have done an exceptional job with this article. There is one suggestion i would like to propose : it would've been great if there was a comparative analysis with some other traditional villages from other cultures and countries. With or without it, you have really shown how a work should be done and u have set an example of real researchers.
Author Response
This paper is a part of Identifying Landscape and Integrated Protection of Traditional Minority National Settlement based on Landscape Character Assessment in Wuling Mountainous Areas supported by NSFC. The relationship between traditional villages studied in this paper is based on the reasons for the formation of settlements. Because the reasons for the formation of settlements in China and other countries are quite different, it is difficult to compare.
Thank you very much for your advice. We will try further in future research.
Reviewer 2 Report
This paper applied the social network analysis (SNA) to construct a traditional village landscape network (TVLN) for the Yuan River Basin in China and evaluated the impact of rapid urbanization on traditional village landscapes from the perspective of network topology. In general, the topic of this study is interesting and has some implications for sustainable management of traditional villages in China. This study suggests that the landscape network of traditional villages in the Yuan River Basin have a dendritic and multi-core structure, but the existence of tangent and isolated points would undermine the stability of the TVLN. Given the paper’s topic significance and research design, this research fits in the main theme of the journal. However, the academic writing should be further improved. There are a few problems including format, the use of the preposition “the”, punctuation, specific writing style, etc., please check and revise accordingly. Below are some specific comments:
Introduction
Line 41, I didn’t find the numbers from the paper cited. Please double check it. “Unincorporated village” should be “natural village”, and “incorporated village” should be “administrative village”. You may also need to update the numbers to the latest as it is year 2021 now.
Methods
- It is necessary that all of the figure/maps must comply with the relevant national, so it is suggested the authors redraw Figure 1 by adding the South China Sea Islands in the map of whole national map.
- In Section 2.2.1, when introducing factors, please clearly indicate the relationship between each factor and the connection strength. Why the land fragmentation index could represent the cultural dimension of connection strength? Does a more fragmented landscape suggest a higher connection strength between traditional villages or lower?
- In Section 2.2.3, why the top 3% of the relationship strengths are selected to form the links of the TVLN, not 5%, 10% or all? The authors claimed in “Conclusion” that 3% of the network density would not change the properties of the network. Why? Please provide evidence.
- Please consider revising the title of Figure 2, I don’t think your research has anything to do with social network, just network topology should be fine.
- Line 150: “the culture factor” should be “the cultural factor”.
Results
- The numbers in Figure 2, 5 and 6 are hard to read.
- The description of TVLN’s characteristics in Section 3.2 is centered on the extracted network. It is suggested the authors present the results focusing more on traditional villages.
Discussion
- Line 303, the definition of “cluster” should be moved to “Materials and Methods” when it first appears.
- Section 4.2, the policy implications should be more specific. For example, Line 299-300, what specific protection plans or strategies are needed for those tangent and isolated points?
Author Response
Thanks for your time and carefully reading. We summarize the respond as following.
|
Line 41, I didn’t find the numbers from the paper cited. |
We revised the references. “Hu Binbin, Li Xiangjun, Wang Xiaobo. Blue Book of Chinese Traditional Villages: Investigation Report on the Projection of Chinese Traditional Villages [M]. Social science academic press (China)”. |
|
Please check it. “Unincorporated village” should be “natural village”, and “incorporated village” should be “administrative village”. You may also need to update the numbers to the latest as it is year 2021 now. |
We replaced “Unincorporated village” and “incorporated village” with “natural village” and “administrative village”. |
|
It is necessary that all the figure/maps must comply with the relevant national, so it is suggested the authors redraw Figure 1 by adding the South China Sea Islands in the map of whole national map. |
We redrawn figure 1 and added the South China Sea Islands in Chinese map. |
|
Please consider revising the title of Figure 2, I don’t think your research has anything to do with social network, just network topology should be fine. |
We removed Figure 2 and reedited the relevant parts of Figure 2. |
|
Line 150: “the culture factor” should be “the cultural factor”. |
We corrected the grammatical errors. |
|
The numbers in Figure 2, 5 and 6 are hard to read. |
These two figures are generated by software, and the numbers only represent the node number. If we edit the pictures, they will lose their authenticity. |
|
The description of TVLN’s characteristics in Section 3.2 is centered on the extracted network. It is suggested the authors present the results focusing more on traditional villages. |
We want to highlight the logical connection between traditional village nodes in this paper. Therefore, the topological relationship between nodes is analyzed. |
|
Line 303, the definition of “cluster” should be moved to “Materials and Methods” when it first appears. |
“Cluster” is not the core research method of this paper, but a description of a traditional village protection model. Therefore, we don't think it is necessary to discuss it in the method part. |
|
Section 4.2, the policy implications should be more specific. For example, Line 299-300, what specific protection plans or strategies are needed for those tangent and isolated points? |
We further give special suggestions on the protection measures of outliers. |
Reviewer 3 Report
The strengths of this manuscript are:
- The topic is very actual, specially in China
- the existence of survey research and the fact that the analysis and the findings result from this.
-As the title of the article is 'Traditional village landscapes,' I consider that the issue of the 'Traditional Villages in general and specially in China should be more detailed. Therefore, this topic requests additional background and literature review.
-Keywords: please the author not insert the same words of the title.
-The review literature of traditional village landscape(paragraph introduction) appear not sufficient. Also the importance of networks on territory. Maybe the author could talk in more depth about what this paragraph is trying to say. It would help to understand the argument more clearly and methods applied. There is a rich comprehensive academic literature. -The methods applied are appropriate. -
At the end a small discussion section is made to detail the objectives of the study. Link back to the introduction by summarizing what the author aimed to find out and overall how this manuscript has contributed – in one succinct sentence.
A option would be to provide a more critical discussion on the issues.
In your Discussion section discuss your findings in relation to the literature you have reported in the beginning. Here you have to construct your contribution. In what way do you inform existing knowledge related to the use of the TVLN. -The audience of this paper can range from the wide public to a more specialized one. Probably, in the conclusion the impact for the territory would be more schematic and inserted. -The results of the SNA are presented clearly, without much of a discussion. The discussion and concluding remarks are quite generic.- Implications for research, practice and/or society: Theoretical and managerial implications are not quite clearly identified.
-The manuscript is duly structured in terms of information provided and clarity of expression less of continue readability. However there is an effort of originality.
Author Response
Thanks very much for your careful reading and thougtful comments. We summarized the respond as following.
|
Major comments |
Mirror comment: |
|
As the title of the article is 'Traditional village landscapes,' I consider that the issue of the 'Traditional Villages in general and specially in China should be more detailed. Therefore, this topic requests additional background and literature review. |
We further elaborate on Chinese traditional villages in lines 52-62. |
|
Keywords: please the author not insert the same words of the title. |
We modified the keywords. Landscape heritage; connection strength; social network analysis; rural landscape; traditional village; sustainable protection |
|
Link back to the introduction by summarizing what the author aimed to find out and overall how this manuscript has contributed – in one succinct sentence. |
In line 365, we revised and reviewed the theme of this paper: an evaluation method of the relationship between traditional settlements. |
|
Implications for research, practice and/or society: Theoretical and managerial implications are not quite clearly identified. |
We put forward specific suggestions on the management and protection of isolated points in line 317. |
Round 2
Reviewer 2 Report
The authors did not attend to my second question "You may also need to update the numbers to the latest as it is year 2021 now."
Author Response
Thank you very much for your careful reading and pointing out the shortcomings of the paper. After modification, this paper has more timeliness.
Q:The authors did not attend to my second question "You may also need to update the numbers to the latest as it is year 2021 now."
We have updated the data to China's villages data in 2020 newly released by the China Bureau of statistics.
Reviewer 3 Report
The article is improved.
Author Response
Thank you very much for your careful reading and pointing out the shortcomings of the paper. After modification, this paper is more rigorous.